# Minimized human telomerase maintains telomeres and resolves endogenous roles of H/ACA proteins, TCAB1, and Cajal bodies

Jacob M Vogan, Xiaozhu Zhang, Daniel T Youmans[†], Samuel G Regalado, Joshua Z Johnson, Dirk Hockemeyer, Kathleen Collins*

Department of Molecular and Cell Biology, University of California, Berkeley, Berkeley, United States

**Abstract** We dissected the importance of human telomerase biogenesis and trafficking pathways for telomere maintenance. Biological stability of human telomerase RNA (hTR) relies on H/ACA proteins, but other eukaryotes use other RNP assembly pathways. To investigate additional rationale for human telomerase assembly as H/ACA RNP, we developed a minimized cellular hTR. Remarkably, with only binding sites for telomerase reverse transcriptase (TERT), minimized hTR assembled biologically active enzyme. TERT overexpression was required for cellular interaction with minimized hTR, indicating that H/ACA RNP assembly enhances endogenous hTR-TERT interaction. Telomere maintenance by minimized telomerase was unaffected by the elimination of the telomerase holoenzyme Cajal body chaperone TCAB1 or the Cajal body scaffold protein Coilin. Surprisingly, wild-type hTR also maintained and elongated telomeres in TCAB1 or Coilin knockout cells, with distinct changes in telomerase action. Overall, we elucidate trafficking requirements for telomerase biogenesis and function and expand mechanisms by which altered telomere maintenance engenders human disease.

*For correspondence: kcollins@berkeley.edu

Present address: †University of Colorado, Aurora, United States

## Introduction

Eukaryotic chromosome stability relies on end-protective telomeres (*Arnoult and Karlseder, 2015*). In most organisms, end-protective telomeric chromatin assembles on a tract of simple-sequence DNA repeats (*de Lange, 2010*), for example the human repeat of TTAGGG on the strand with 3'OH terminus. Maintenance of this telomeric repeat array requires *de novo* repeat synthesis by the ribonucleoprotein (RNP) reverse transcriptase telomerase to balance the repeat erosion inherent in DNA-dependent DNA-polymerase replication of the genome (*Blackburn et al., 2006*; *Hug and Lingner, 2006*). Telomerase extends chromosome 3' ends by copying a template within the telomerase RNA subunit (hTR in human cells), using an active site in the telomerase reverse transcriptase protein (TERT). The intricate co-folding and co-function of telomerase RNA and TERT obliges a step-wise RNP assembly process and generates a network of protein- and RNA-domain interactions (*Blackburn and Collins, 2011*, *Schmidt and Cech, 2015*).

Cellular RNP biogenesis involves transit through and concentration in specific nuclear bodies (*Mao et al., 2011*; *Machyna et al., 2013*). Trafficking pathways differ depending on the diverse steps of RNA processing, modification, and RNP assembly that give a transcript its fate and function. Among the best-studied RNP transit points are Cajal bodies, defined as foci of the protein Coilin (*Nizami et al., 2010*; *Machyna et al., 2015*). Enzymes resident in Cajal bodies catalyze numerous RNA processing and modification reactions as well as RNP assembly and remodeling

**eLife digest** Most cells in the human body can only divide a certain number of times before they die. This is because regions called telomeres at the ends of the cell's DNA get shorter every time the cell divides, to the point that they disappear and halt cell growth. Particular types of cells – including some stem cells and cancer cells – can avoid death and continue to divide indefinitely because they produce an enzyme called telomerase that extends the telomere regions.

The process by which the telomerase enzyme binds to and lengthens the DNA has several stages and involves many different proteins. One of the stages involves moving telomerase from the sites where it is assembled within the cell to a place where it can find telomeres in need of elongation (different areas within the cell compartment called the nucleus). Structures inside the nucleus called Cajal bodies were thought to help the enzyme bind to the telomeres. It is not clear why the process of extending telomeres is so complex.

Vogan et al. engineered altered versions of telomerase that use simpler pathways to bind to and act on telomeres and inserted them into 'pluripotent' stem cells and cancer cells from humans. The experiments show that a pathway that helps to move the enzyme from its normal storage place in the nucleus is less important for extending telomeres in cancer cells than in pluripotent stem cells. Unexpectedly, Cajal bodies are not critical for bringing telomerase into contact with the telomeres in either cell type.

The findings show that many of the proteins involved in extending telomeres in cells are not strictly essential. The simplified pathway developed by Vogan et al. opens up new opportunities to study the details of how telomerase extends telomeres.

(*Machyna et al., 2013*). Beyond RNA processing and RNP biogenesis factors, Cajal bodies also recruit regulatory complexes such as CDK2-cyclinE (*Liu et al., 2000*) and have widespread influence on gene expression (*Wang et al., 2016*). Despite the multiplicity of functions ascribed to Cajal bodies, including critical roles in vertebrate telomerase function described below, it remains unclear whether their formation is a cause or consequence of associated RNP biogenesis pathways.

Curiously, ciliate, fungal, and vertebrate telomerases follow entirely different RNP biogenesis pathways, which are directed by telomerase RNA interaction with a La-motif protein, Sm proteins, or H/ACA proteins, respectively (*Egan and Collins, 2012a*). In human cells, telomerase shares the same mature H/ACA proteins (dyskerin, NHP2, NOP10, GAR1) and H/ACA RNP biogenesis chaperones as the intron-encoded small nucleolar (sno) or small Cajal body (sca) RNPs that catalyze cleavage and pseudouridylation of ribosomal and small nuclear RNAs (*Kiss et al., 2010*). Because precursor hTR is released from its site of synthesis as an autonomous transcript rather than the spliced intron lariat of other human H/ACA RNAs, it is sensitized to degradation in dyskeratosis congenita (DC) patient cells with a mutation of an H/ACA protein (*Egan and Collins, 2012b*; *Armanios and Blackburn, 2012*; *Sarek et al., 2015*). Also, unlike other H/ACA RNAs, hTR requires a 5' trimethylguanosine cap to prevent 5'-3' exonuclease processing (*Mitchell et al., 1999*). Models for vertebrate telomerase RNA trafficking suggest an initial transit of Cajal bodies, where 5' trimethylguanosine cap modification is thought to occur, followed by localization to nucleoli (*Egan and Collins, 2012a*). Subsequent RNP trafficking from nucleoli to steady-state concentration in Cajal bodies depends on the binding of the Cajal body chaperone and telomerase holoenzyme protein TCAB1/WDR79/WRAP53β to an hTR 3' stem-loop CAB-box motif (*Venteicher et al., 2009*; *Tycowski et al., 2009*; *Zhong et al., 2011*), which is present in both stem-loops of an H/ACA scaRNA (*Kiss et al., 2010*). Overall, this trafficking complexity could represent only a subset of the necessary cellular directions for human telomerase biogenesis and function.

The human telomerase holoenzyme subunits that localize active RNP to Cajal bodies are considered crucial for telomerase action at telomeres (*Schmidt and Cech, 2015*). Transient telomere colocalization with a Cajal body can be detected in S-phase, when telomerase acts at chromosome ends (*Jády et al., 2006*; *Tomlinson et al., 2006*). Evidence for Cajal body delivery of telomerase to telomeres builds from studies depleting TCAB1 or Coilin using RNA interference, which reduced or eliminated hTR colocalization with telomeres and induced telomere shortening (*Venteicher et al., 2009*;

*Zhong et al., 2011*; *Stern et al., 2012*; *Zhong et al., 2012*). DC patient cells with biallelic TCAB1 mutations have short telomeres and fail to maintain telomere length even with the up-regulated telomerase expression in induced pluripotent cells (*Batista et al., 2011*). With all of this experimental support for Cajal body delivery of human telomerase to telomeres, it is puzzling that mouse telomerase RNA does not localize to Cajal bodies (*Tomlinson et al., 2010*). Also, recently, Coilin gene knock-out (KO) in HeLa cells generated two clonal cell lines that maintained telomeres (*Chen et al., 2015*).

Here, we address the significance of human telomerase biogenesis and trafficking pathways for active RNP assembly and function at telomeres. As a new approach, we first bypassed the endogenous hTR stability requirement for H/ACA proteins. Remarkably, we found that a minimal hTR (hTRmin) containing only binding sites for TERT can assemble active RNP in cells, and this active RNP can maintain stable telomere length homeostasis. TERT overexpression was required for cellular assembly with hTRmin, suggesting that hTR H/ACA RNP assembly enhances TERT interaction at scarce endogenous telomerase subunit levels. KO of TCAB1 or Coilin did not alter hTRmin RNP function at telomeres. Surprisingly, TCAB1 KO or Coilin KO was also permissive for hTR telomerase to maintain telomeres. In cancer and pluripotent stem cell lines with endogenous hTR and TERT expression, TCAB1 KO resulted in a slow decline of telomere length followed by stable telomere length homeostasis. We conclude that aside from conferring hTR stability and facilitating active telomerase assembly in cells with scarce TERT, hTR assembly as H/ACA RNP is not essential. Also, Cajal body localization is not essential for hTRmin or hTR telomerase to maintain stable telomere length homeostasis. Our findings illuminate the influences of nuclear trafficking on human telomerase biogenesis and action at telomeres, account for why telomerase biogenesis pathways can be so divergent in eukaryotic evolution, and give new interpretation to the mechanisms by which different telomerase subunit mutations impose human disease.

## Results

### Human telomerase can be liberated from H/ACA RNP assembly

To test the significance of H/ACA RNP biogenesis for human telomerase function at telomeres, we sought to bypass the essential role of this pathway in the protection of hTR from degradation while preserving hTR-TERT interaction and RNP catalytic activity. The H/ACA-motif 5′ stem of hTR (*Figure 1A*) separates two hTR regions that are critical for binding of TERT and catalytic activity: the template/pseudoknot (t/PK) and conserved regions (CR) 4/5 (*Zhang et al., 2011*). Previously we joined these two regions with a spacer to create hTRmin (*Figure 1B*), which in combination with TERT reconstitutes a minimized active telomerase in rabbit reticulocyte lysate (*Wu and Collins, 2014*). To allow cellular accumulation of hTRmin, the most successful strategy was to append 3′ processing and protection motifs from the human long non-coding RNA MALAT1 (*Brown et al., 2012*; *2014*).

The overall design strategy to optimize hTRmin accumulation involved testing combinations and variations of RNA modules (*Figure 1C*). For example, we interchanged the hTRmin 5′ end as no leader, endogenous hTR quadruplex-forming leader (LeaderG), or a mutant leader (LeaderC) that eliminates one of the four adjacent guanosine tracts (*Sexton and Collins, 2011*). We also tested 6 or 14 nucleotide (nt) lengths of single-stranded RNA spacer separating the t/PK and CR4/5 regions. For 3′-end formation, we tested the MALAT1 triplex motif with or without an adjacent RNase P cleavage site mimicking a pre-tRNA (*Brown et al., 2012*). We also omitted or included a hepatitis delta virus ribozyme (HDV RZ), which can increase hTR accumulation by stabilizing the precursor 3′ end with an exonuclease-resistant 2′,3′ cyclic phosphate (*Egan and Collins, 2012b*).

We first assessed hTRmin accumulation in transiently transfected 293T cells, using an expression context previously optimized for hTR that exploits the U3 snoRNA promoter to direct transcription by RNA Polymerase II (*Fu and Collins, 2003*). Accumulation of hTRmin to a level detectable by Northern blot required both the RNA triplex and the RNase P site (*Figure 1D*). We next tested if this cellular hTRmin could form active RNP. We transiently co-expressed hTRmin or hTR with 3xFLAG-tagged (F) TERT in the telomerase-negative U2OS cell line and assayed for telomeric primer extension by immunopurified TERT. Cellular reconstitution with hTRmin yielded lower RNA accumulation and proportionally lower overall activity than reconstitution with hTR, but hTRmin and hTR

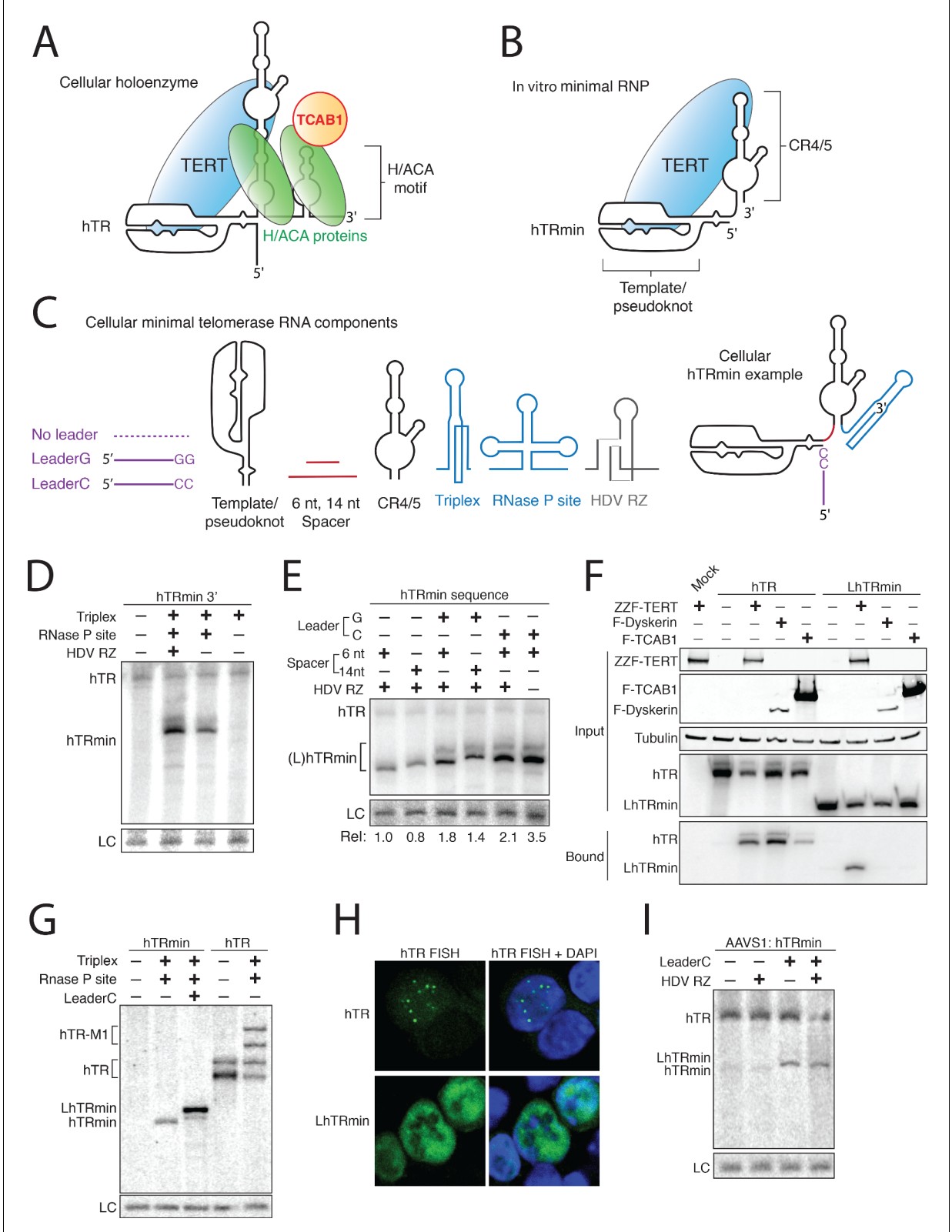

**Figure 1.** Human telomerase RNA can accumulate without H/ACA RNP biogenesis. (**A,B**) Diagrams of hTR and hTRmin secondary structure and bound proteins. (**C**) Parts list for a cellular minimal telomerase RNA. Components are presented in 5' to 3' order. (**D,E**) Northern blot assay for RNA accumulation in transfected 293T cells. Endogenous hTR was also detected. Loading control (LC) is a cellular RNA non-specifically detected by the Northern blot probe used for normalization. In (**D**), all hTRmin variants are without a 5' leader and with a 6 nt spacer. In (**E**), all constructs had the RNA

*Figure 1 continued on next page*

*Figure 1 continued*

triplex and RNase P site. hTRmin accumulation was normalized to LC to quantify relative accumulation (Rel). (**F**) Copurification of hTR or LhTRmin with tagged telomerase holoenzyme subunits co-overexpressed in transfected VA-13 cells. RNPs were purified from cell lysate using FLAG antibody resin and analyzed by immunoblot and Northern blot. F indicates 3xFLAG peptide, ZZ indicates tandem Protein A domains. (**G**) Northern blot assay for RNA accumulation in transfected VA-13 cells. RNA folding during extensive gel electrophoresis gives mature hTR two mobilities (a doublet of bands). (**H**) FISH detection of hTR or LhTRmin in transfected HCT116 cells. Untagged TERT was coexpressed. (**I**) Northern blot assay for RNA expressed from transgenes integrated at; in HCT116 cells. Endogenous hTR was also detected.

The following figure supplement is available for figure 1:

**Figure supplement 1.** Cellular assembly of hTRmin telomerase.

telomerase RNPs had similar profiles of repeat synthesis (*Figure 1—figure supplement 1A*). For continued optimization, we next compared hTRmin accumulation with no 5' leader, LeaderG, or LeaderC and with 6 nt versus 14 nt spacer (*Figure 1C*), all with the RNA triplex and RNaseP cleavage site. The presence of a 5' leader increased accumulation, and the quadruplex-mutant LeaderC was as good or better than the native hTR LeaderG (*Figure 1E*). Also, the 6 nt spacer between t/PK and CR4/5 was as good or better than the 14 nt spacer (*Figure 1E*). HDV RZ addition downstream of the MALAT1 3' processing elements improved hTRmin accumulation from some transfected constructs but did not notably increase accumulation of the optimal LeaderC-containing hTRmin with 6 nt spacer (*Figure 1E*), which in subsequent experiments we designate LhTRmin for distinction from the leader-less hTRmin.

To confirm that removal of the H/ACA motif eliminated all interactions with H/ACA proteins, we expressed epitope-tagged TERT, dyskerin, or TCAB1 with hTR or LhTRmin in telomerase-negative VA-13 cells, immunopurified the tagged protein using FLAG antibody resin, and detected bound RNA by Northern blot (*Figure 1F*). As expected, TERT, dyskerin, and TCAB1 each bound hTR whereas only TERT bound LhTRmin. As in 293T cells, in VA-13 cells both leader-free hTRmin and LhTRmin accumulated, with LhTRmin being optimal (*Figure 1G*). The robust 3' end protection activity of the MALAT1 triplex was evident when it was appended to full-length hTR (*Figure 1G*; note that mature hTR sometimes migrates as a doublet due to in-gel folding). To compare the cellular localizations of LhTRmin and hTR, we overexpressed LhTRmin or hTR with TERT by transient transfection. While hTR concentrated in nuclear puncta as expected, LhTRmin distributed throughout the nucleoplasm (*Figure 1H*). Transfection of cells lacking endogenous hTR (see below) confirmed that hTRmin and LhTRmin have a diffuse nucleoplasmic distribution distinct from that of hTR or hTR with a CAB-box mutation (*Figure 1—figure supplement 1B*). Together the studies above establish that cellular hTRmin and LhTRmin escape the confines of H/ACA RNP assembly.

We next tested whether hTRmin or LhTRmin could accumulate when stably expressed from an integrated transgene. In the near-diploid HCT116 human colon carcinoma cell line, we used a zinc finger nuclease that introduced a double-stranded DNA break at the AAVS1 safe-harbor locus (*Hockemeyer et al., 2009*) to integrate an RNA expression cassette and a neomycin resistance cassette. After selection, the polyclonal population of targeted cells was assayed for RNA accumulation by Northern blot. Consistent with results from transient transfection, LhTRmin accumulation exceeded that of hTRmin, with or without a downstream HDV RZ (*Figure 1I*). Stably expressed hTRmin was barely detectable, and even LhTRmin accumulated to a level lower than endogenous hTR (*Figure 1I*). Therefore, the MALAT1 3' processing elements do not confer as much accumulation to minimized hTR as does an embedded H/ACA RNP.

## Catalytically active telomerase supports telomere maintenance without H/ACA proteins or TCAB1

Successful cellular biogenesis of hTRmin telomerase RNP allowed us to test whether H/ACA proteins have post-biogenesis influences on telomerase function at telomeres. We first programmed Cas9 for cleavage of the endogenous hTR locus (TERC) in HCT116 cells in the presence of donor plasmid that would integrate a puromycin resistance cassette (*Figure 2A*). After selection, clonal cell lines were established and screened by PCR for biallelic gene disruption (*Figure 2—figure supplement 1A*). Two independent cell lines with homozygous TERC KO were confirmed to lack telomerase catalytic

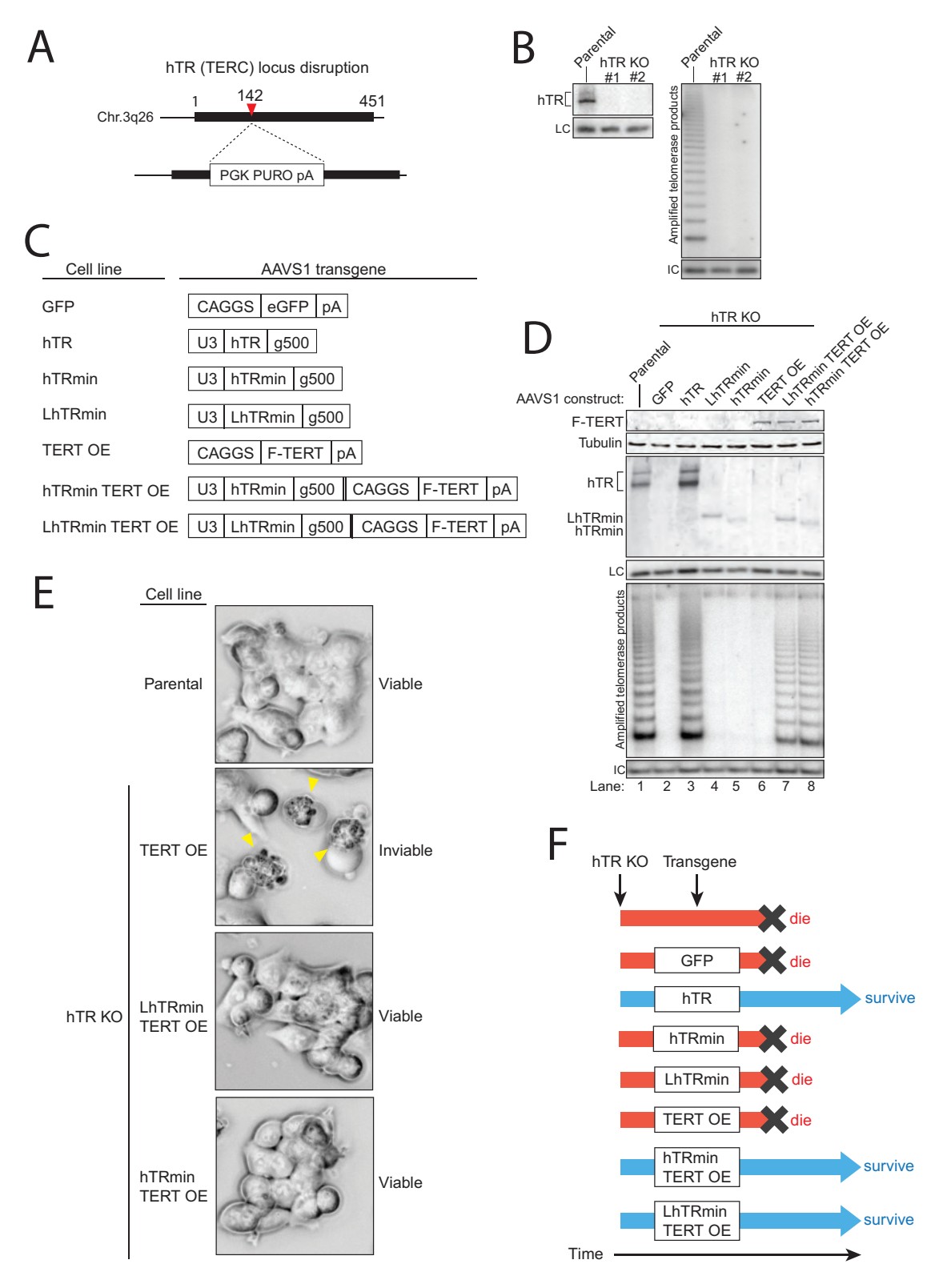

**Figure 2.** An hTRmin RNP can functionally substitute for hTR. (**A**) Schematic of Cas9-mediated disruption of the hTR locus with a PURO selection cassette. (**B**) Northern blot and hotTRAP assays of hTR KO#1 and KO#2 clonal HCT116 cell lines. An internal control (IC) to normalize PCR amplification was always included in hotTRAP assays. (**C**) AAVS1 donor constructs used for transgene rescue of hTR KO HCT116 cells. RNA expression used the U3 promoter and terminated within 500 bp of transplanted genomic region from immediately downstream of endogenous hTR (g500). mRNA expression

*Figure 2 continued on next page*

*Figure 2 continued*

used the CAGGS promoter terminated with a polyadenylation element (pA). (D) Immunoblot, northern blot, and hotTRAP characterization of HCT116 cell lines 51 days after hTR KO targeting. (E) Brightfield microscopy images of HCT116 cell lines during the die-off interval of telomerase-negative cell cultures. Yellow arrowheads indicate membrane blebbing. (F) Chart of survival fate of HCT116 hTR KO cell lines with the indicated transgene at AAVS1.

The following figure supplement is available for figure 2:

**Figure supplement 1.** Generation of cell lines expressing hTRmin telomerase.

activity in cell extract, as assayed by the PCR-based telomeric repeat amplification protocol performed with radiolabeled dGTP (hotTRAP) (*Figure 2B* and *Figure 2—figure supplement 1B*, left). Initially, these two hTR KO clonal cell lines showed no growth defect.

In each hTR KO cell line, we targeted transgene integration at AAVS1 with expression cassette(s) encoding GFP, N-terminally F-tagged TERT, hTR, LhTRmin, hTRmin, or LhTRmin or hTRmin coexpressed with TERT (*Figure 2C*). RNA expression was directed by the U3 snoRNA promoter, and protein expression was driven by the strong and constitutive CAGGS promoter. Polyclonal populations of transgene-containing cells were selected and immediately assayed for telomerase subunit expression and telomerase activity (*Figure 2D*). TERT overexpression (OE) was detected by immunoblot, and transgene-encoded RNA expression was detected by Northern blot. Telomerase catalytic activity was assayed by hotTRAP.

Transgene expression of wild-type hTR restored telomerase catalytic activity to the hTR KO cells, rescuing endogenous TERC locus disruption (*Figure 2D*, lane 3). Transgene expression of GFP, LhTRmin, hTRmin, or TERT alone did not (*Figure 2D*, lanes 2 and 4–6). However, LhTRmin or hTRmin with TERT OE did generate active telomerase (*Figure 2D*, lanes 7–8). TERT co-expression did not affect hTRmin biological accumulation (*Figure 2D*, compare lanes 4–5 to lanes 7–8). Parallel results were confirmed using the independent hTR KO clonal cell line (*Figure 2—figure supplement 1B,C*). In addition, we tested telomerase activity rescue of hTR KO cells or hTR KO cells with AAVS1 TERT OE by lentiviral introduction of LhTRmin. Only with TERT OE did lentiviral expression of LhTRmin produce active telomerase, whereas lentiviral expression of hTR rescued hTR KO without TERT OE (*Figure 2—figure supplement 1D,E*). Telomerase activity levels in the hTRmin telomerase cell lines were within an order of magnitude of the telomerase level in parental HCT116 cells (*Figure 2—figure supplement 1B,C*). From these experiments we conclude that hTR assembly as H/ACA RNP strongly stimulates hTR interaction with TERT when both subunits are at very low expression levels, but this role of H/ACA RNP assembly can be bypassed by increasing the cellular availability of TERT.

The hTR KO cell lines lacking active telomerase ultimately entered an interval of pervasive cell death with dramatic membrane blebbing (*Figure 2E*, yellow arrowheads) at ~70 days post-targeting, corresponding to ~70 population doublings. In stark contrast, all of the telomerase-positive polyclonal cell cultures and clonal cell lines proliferated over many months of continuous passage with normal morphology and doubling time (*Figure 2F*). Therefore, the catalytically active hTRmin telomerase RNPs conferred indefinite cellular proliferative capacity.

We next tested the hTR KO cell lines with AAVS1 transgenes for rescue of telomere shortening. As expected, hTR KO cells re-expressing hTR rapidly gained telomere length, whereas hTR KO cells expressing the negative control GFP did not (*Figure 3A*, lanes 1–3 and *Figure 3—figure supplement 1A*). All cell cultures lacking active telomerase had short telomeres that continued to shorten until eventually all cells in the culture died (*Figure 3A,B*). Telomere length was heterogeneous in the polyclonal population of hTR KO cells rescued by LhTRmin with TERT OE (*Figure 3A*, lane 7 and *Figure 3—figure supplement 1B*). We generated clonal cell lines from the hTR KO cell lines complemented by LhTRmin or hTRmin with TERT OE and characterized their maintenance of telomere length. These clonal cell lines had distinct but stable telomere lengths (*Figure 3C,D* and *Figure 3—figure supplement 1C*).

To investigate the clonal cell line variance in telomere length at length homeostasis, we profiled levels of telomerase subunits and activity across the clonal cell lines. All of the LhTRmin cell lines had higher steady-state RNA accumulation than the hTRmin lines (*Figure 3E*). LhTRmin levels varied more than hTRmin, and TERT levels varied more in LhTRmin lines than hTRmin lines (*Figure 3E* and *Figure 3F*, symbols). Telomerase activity measured by fluorescence quantification of telomeric

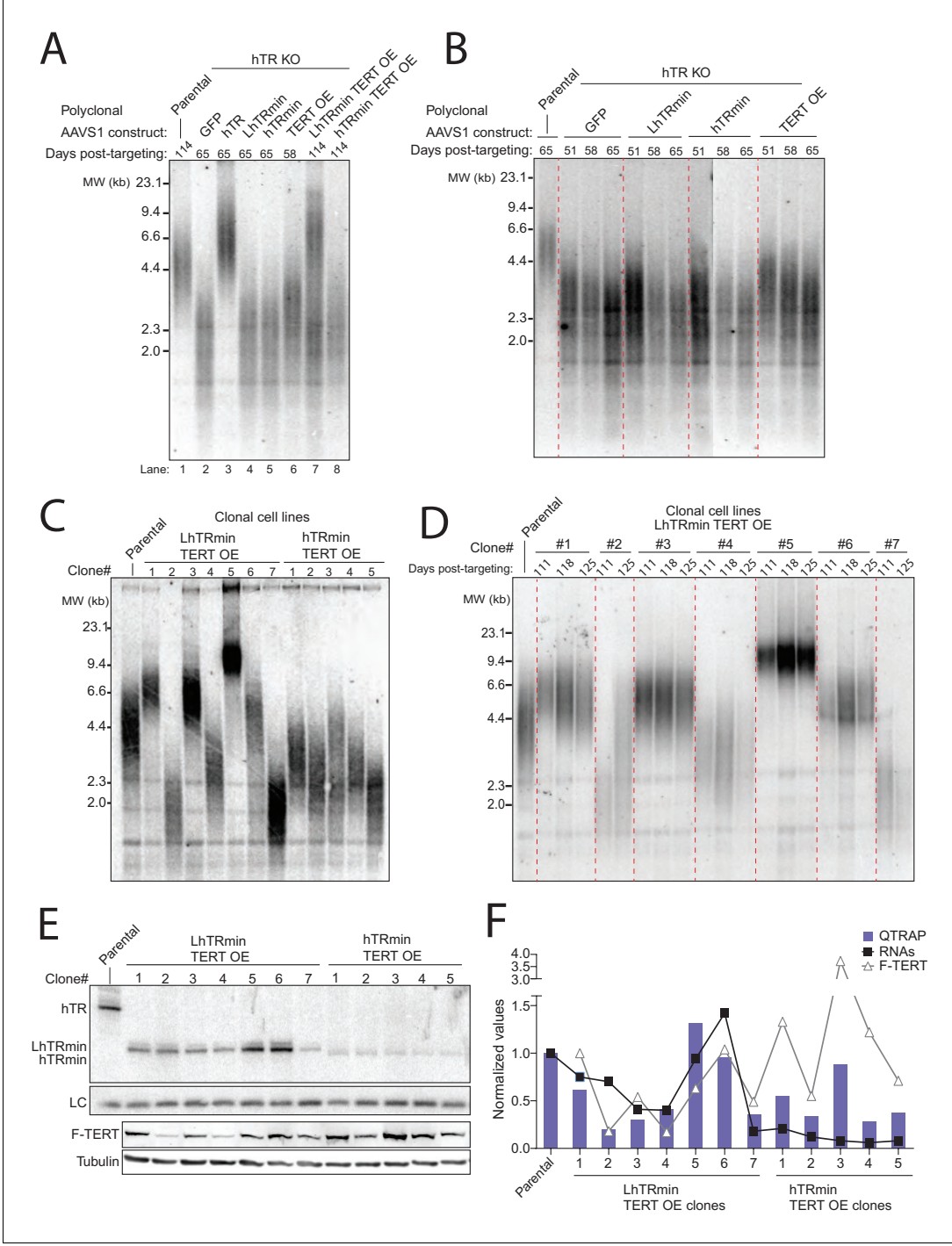

**Figure 3.** Telomerase with hTRmin supports stable telomere length maintenance. (A) Southern blot detection of telomere restriction fragment lengths (TRF) for HCT116 cell lines after release from selection. The telomerase-negative cell line TRFs were analyzed before cultures commenced cell death. (B) Time course of TRF shortening in the telomerase-negative HCT116 cell lines. The 65 days post-targeting time point was the final cell collection before culture death. All lanes are from the same gel. Red dashed lines separate different genotypes. (C) TRF analysis for multiple clonal cell lines expressing LhTRmin with TERT OE or hTRmin with TERT OE, all cultured in parallel and assayed at the same time point 111 days post-targeting. (D) Time course of TRF in the clonal cell lines expressing LhTRmin with TERT OE or hTRmin with TERT OE. Days post-targeting refers to the hTR KO. (E) Immunoblot and Northern blot analysis of telomerase subunit expression levels across the clonal cell lines expressing LhTRmin with TERT OE or hTRmin with TERT OE, performed using cells at 111 days post-targeting. (F)
*Figure 3 continued on next page*

*Figure 3 continued*

Comparison of telomerase activity measured by QTRAP (purple bars) with telomerase subunit expression levels quantified from blots in (E). QTRAP (averaged, n = 5) and RNA signals were normalized to parental HCT116 cell line activity and endogenous hTR. TERT OE signals were set relative to LhTRmin TERT OE clone #1.

The following figure supplement is available for figure 3:

**Figure supplement 1.** Characterization of telomere length maintenance and telomerase activity levels in hTRmin telomerase cell lines.

repeat amplification (QTRAP) (*Figure 3F*, bars and *Figure 3—figure supplement 1D* for replicates) or hotTRAP (*Figure 3—figure supplement 1E*) was generally greatest in cell lines with high telomerase subunit expression (*Figure 3F*, compare symbols to bars). Higher telomerase activity in cell extract correlated generally but not absolutely to longer telomere length at homeostasis (compare *Figure 3F*, bars to *Figure 3C*). We suggest that differences in telomerase subunit expression levels could in part reflect epigenetic differences introduced upon transgene integration, which was an independent event in each clonal cell line. In addition, differences in subunit expression levels could result from the stochastic fluctuation established to occur in many model systems, including HeLa cells (*Bryan et al., 1998*).

## TCAB1 and Cajal bodies contribute non-essential regulations of telomerase

We confirmed that the hTRmin telomerase RNP assembled in the stable cell lines did not have the endogenous hTR RNP interaction with TCAB1, tested by immunopurification of TCAB1 from cell lysate and subsequent telomerase activity assay (*Figure 4—figure supplement 1A*). However, because telomere-associated telomerase would be a small fraction of the total telomerase RNP pool, we sought another approach to demonstrate that hTRmin telomerase maintained telomeres without a requirement for TCAB1-mediated recruitment to Cajal bodies. To this end, we disrupted the genes encoding TCAB1 and Coilin in HCT116 cells expressing hTRmin telomerase.

We programmed Cas9 for cleavage of the endogenous TCAB1 or Coilin locus in the presence of donor plasmid that would integrate a hygromycin resistance cassette (*Figure 4A*). Targeting and selection were highly efficient, resulting in polyclonal populations of LhTRmin TERT OE cells and hTRmin TERT OE cells with little TCAB1 or Coilin, as detected by immunoblot (*Figure 4B*) or immunofluorescence (*Figure 4C*). TCAB1 KO cells retained a normal level of Coilin, and Coilin KO cells retained a normal level of TCAB1 (*Figure 4B*). Cells lacking TCAB1 retained Cajal bodies (*Figure 4C*), in agreement with some previous findings (*Venteicher et al., 2009*; *Zhong et al., 2011*) but contrary to others (*Mahmoudi et al., 2010*; *Wang et al., 2016*). Also, nuclear foci of SMN remained detectable in the TCAB1 KO cells but not Coilin KO cells (*Figure 4—figure supplement 1B*). We conclude that in HCT116 cells, TCAB1 KO did not disrupt Cajal bodies, and Coilin KO did not induce TCAB1 degradation.

We isolated clonal cell lines from the polyclonal KO cell populations and validated homozygous TCAB1 or Coilin KO by genomic locus PCR and protein immunoblots (*Figure 4D* and *Figure 4—figure supplement 1C,D*). These clonal cell lines retained telomerase catalytic activity in cell extract (*Figure 4E*) and stably maintained telomeres (*Figure 4F* and *Figure 4—figure supplement 1E*). Some heterogeneity was evident comparing the telomere lengths maintained in different clonal cell lines, which could result from stochastic variation (*Bryan et al., 1998*). Importantly, neither TCAB1 KO nor Coilin KO affected cell viability, morphology, or proliferation in a readily detectable manner. These findings support the conclusion that an RNP of minimized hTRmin and TERT functions at telomeres without dependence on H/ACA RNP biogenesis or localization pathways.

In parallel, we generated TCAB1, Coilin, and TERT KO HCT116 cells with endogenous hTR expression using Cas9 for TCAB1 and Coilin or a zinc finger nuclease developed previously for TERT (*Sexton et al., 2014*). Clonal cell lines with homozygous KO were identified using genomic locus PCR (*Figure 4—figure supplement 1C,D* and *Figure 5—figure supplement 1A*) and validated for loss of TCAB1 or Coilin but not hTR (*Figure 5A,B*). TERT KO was validated by loss of telomerase activity from cell extract (*Figure 5C*). As expected, TCAB1 KO cells retained Coilin, and Coilin KO

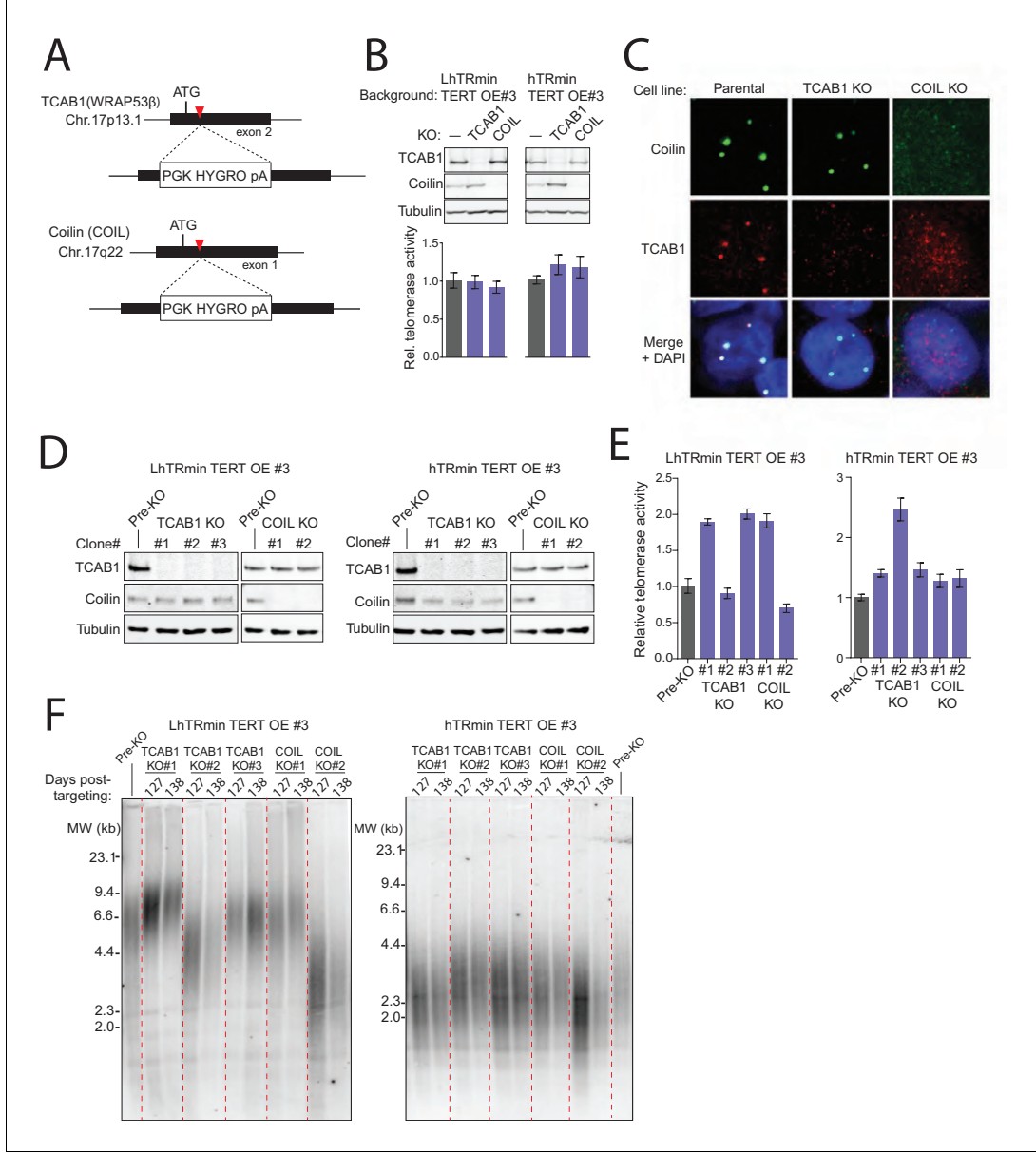

**Figure 4.** TCAB1 and Cajal bodies are not required for telomere maintenance by hTRmin telomerase. (**A**) Schematic of Cas9-mediated disruption of TCAB1 and Coilin (COIL) loci with a HYGRO selection cassette. (**B**) Immunoblot and QTRAP analysis of the polyclonal populations of LhTRmin or hTRmin TERT OE cells selected for disruption of TCAB1 or COIL loci. QTRAP values were normalized to the cell line before TCAB1 or COIL disruption (n = 3). (**C**) Immunofluorescence localization of TCAB1 and Coilin in HCT116 cells. (**D**) Immunoblot analysis for TCAB1 and Coilin in clonal KO cell lines with LhTRmin + TERT OE or hTRmin + TERT OE. (**E**) QTRAP assay of the clonal cell lines in (**D**). QTRAP values were normalized to the cell line before TCAB1 or COIL disruption (n = 3). (**F**) Stable TRF lengths in clonal cell lines lacking TCAB1 or Coilin. Days post-targeting refers to the TCAB1 or COIL KO.

The following figure supplement is available for figure 4:

**Figure supplement 1.** Characterization of HCT116 hTRmin with TERT OE cell lines with TCAB1 KO or Coilin KO.

cells retained TCAB1 (*Figure 5A*). We also confirmed that immunopurification of TCAB1 from Coilin KO cell extract, but not from TCAB1 KO cell extract, enriched active telomerase (*Figure 5—figure supplement 1B*). The loss of TCAB1 or Coilin did not alter telomerase activity in cell extract by more

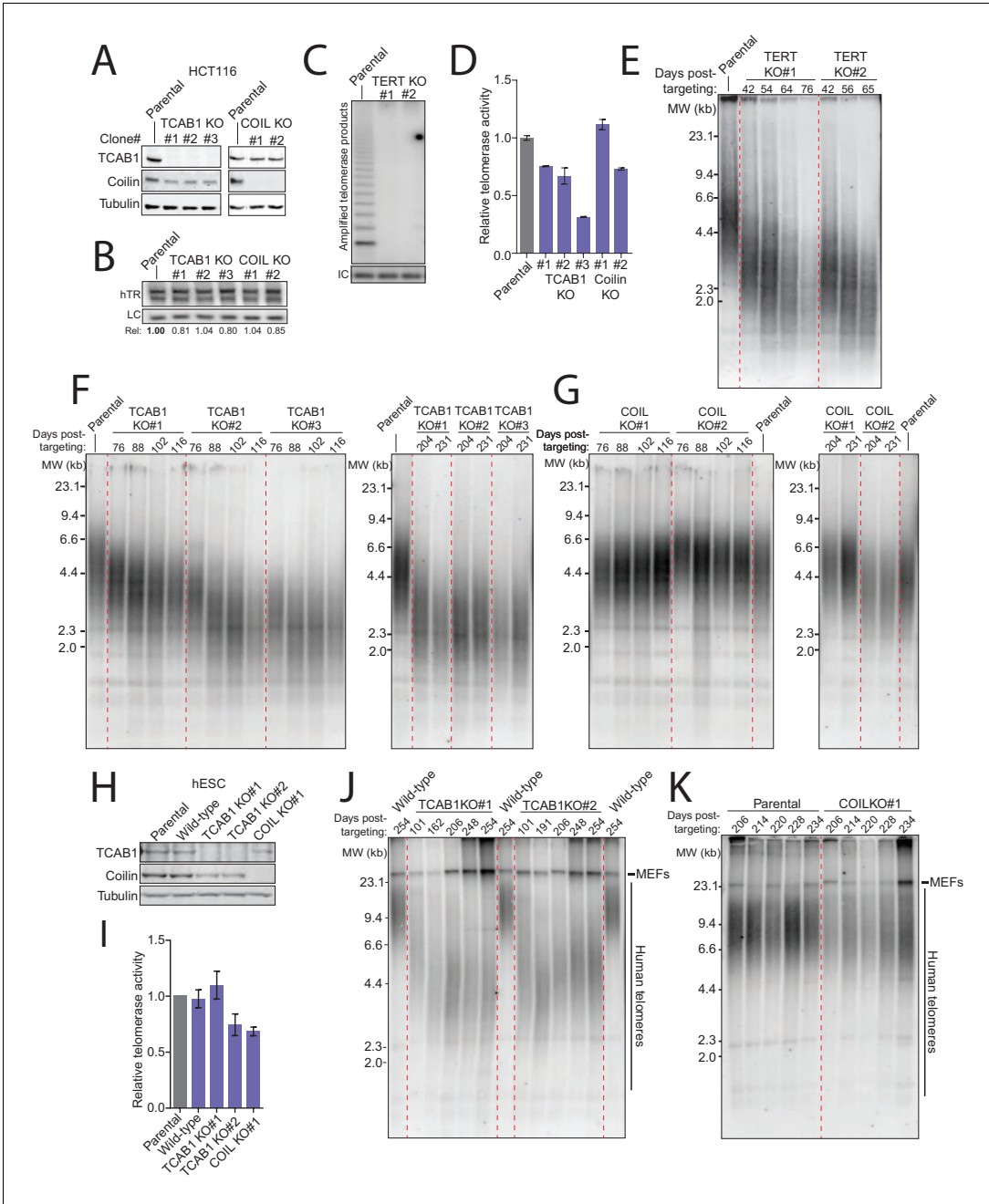

**Figure 5.** TCAB1 and Cajal bodies are not essential for telomere maintenance by endogenous telomerase. (**A**) Immunoblot analysis of HCT116 TCAB1 KO and COIL KO clonal cell lines. (**B**) Northern blot for hTR in HCT116 TCAB1 KO and COIL KO clonal cell lines. (**C**) Lack of hotTRAP telomerase activity detection in the HCT116 TERT KO clonal cell lines. (**D**) QTRAP analysis of telomerase activity in the HCT116 TCAB1 KO and COIL KO clonal cell lines. Values were normalized to the parental HCT116 cell line (n = 3). (**E**) Time course of TRF in the HCT116 TERT KO clonal cell lines. (**F,G**) Time course of TRF in the HCT116 TCAB1 and COIL KO clonal cell lines. (**H**) Immunoblot analysis of hESC TCAB1 KO and COIL KO clonal cell lines. Wild-type refers to an hESC clonal cell line subjected to Cas9 electroporation but retaining a wild-type genotype. (**I**) QTRAP analysis of telomerase activity in the hESC TCAB1 KO and COIL KO clonal cell lines. Values were normalized to the parental hESC line (n = 3). (**J,K**) Time course of TRF in hESC TCAB1 KO and COIL KO clonal cell lines. Note that the long telomeres in mouse cells from the hESC feeder layer contribute some blot signal (indicated MEFs).

The following figure supplements are available for figure 5:

*Figure 5 continued on next page*

*Figure 5 continued*

**Figure supplement 1.** Characterization of HCT116 TERT KO, TCAB1 KO, and COIL KO cell lines with endogenous hTR.
**Figure supplement 2.** Characterization of hESC TCAB1 KO and COIL KO lines with endogenous hTR.
**Figure supplement 3.** Additional controls for TCAB1 loss-of-function in TCAB1 KO cell lines.

than a few-fold extent that could be within the range of clonal variation (*Figure 5D* and *Figure 5—figure supplement 1C*). HCT116 cells with TCAB1 KO or Coilin KO showed no change in growth rate, cell viability, or morphology over more than half a year of continuous culture. In contrast, the TERT KO cells underwent cell death at ~70 days after targeting, paralleling the fate of the hTR KO HCT116 cells.

As expected the TERT KO cells experienced rapid, progressive telomere attrition (*Figure 5E*). Surprisingly, telomeres shortened more gradually in the TCAB1 KO clonal cell lines, followed by stable telomere length homeostasis (*Figure 5F*). The starting point of telomere length in TCAB1 KO clonal cell lines mirrored the amount of telomerase activity in cell extract (compare *Figure 5D* and *Figure 5F*, left panel), but after more than half a year of continuous growth, telomere lengths in all of the TCAB1 KO clonal cell lines stabilized at a few kbp shorter than telomeres in the parental cell line (*Figure 5F*, right panel). In comparison, telomeres in Coilin KO clonal cell lines cultured in parallel remained near the length of telomeres in the parental HCT116 cells (*Figure 5G*). We conclude that in HCT116 cells, neither TCAB1 nor Coilin is required for a stable telomere length homeostasis.

Stable telomere length maintenance in the TCAB1 KO cells with endogenous telomerase was unexpected. To determine whether this finding is general, we investigated the consequence of TCAB1 or Coilin KO in the human embryonic stem cell (hESC) line WIBR#3 (*Chiba and Hockemeyer, 2015*). Clonal hESC lines with TCAB1 or Coilin KO were generated using the same approach that was successful in HCT116 cells (*Figure 5—figure supplement 2A*). As observed for HCT116 cells, hESC lines lacking TCAB1 or Coilin were viable with no evident change in cell morphology or proliferation. TCAB1 accumulated in Coilin KO cells, and Coilin accumulated in TCAB1 KO cells (*Figure 5H*). Neither TCAB1 KO nor Coilin KO affected telomerase activity assayed in cell extract beyond the range of clonal variation (*Figure 5I* and *Figure 5—figure supplement 2B*). Coilin remained localized to Cajal bodies in TCAB1 KO cells (*Figure 5—figure supplement 2C*). Over many months of continuous culture, hESCs lacking TCAB1 experienced very gradual telomere shortening followed by telomere length maintenance (*Figure 5J*). The rate of telomere shortening was slow compared to telomere shortening in TERT KO hESC (*Sexton et al., 2014*). The hESCs lacking Coilin retained telomere lengths comparable to the parental hESC culture (*Figure 5K*). Clonal hESC lines that had undergone targeting but retained the wild-type genotype also did not demonstrate telomere shortening (*Figure 5J*, lanes labeled wild-type).

To confirm that telomere shortening in TCAB1 KO cells was directly linked to the loss of TCAB1, we introduced a TCAB1 transgene to TCAB1 KO cells by integration at AAVS1. TCAB1 KO HCT116 cells ectopically expressing F-tagged TCAB1 regained telomere length (*Figure 5—figure supplement 3A,B*). Likewise, TCAB1 KO hESCs ectopically expressing F-tagged TCAB1 regained telomere length (*Figure 5—figure supplement 3C,D*). As a control for complete TCAB1 loss-of-function, we targeted the TCAB1 KO cells above, which have an exon 2 disruption shortly after the translation start codon (*Figure 4A*), for successful homozygous disruption of downstream exon 8 (*Figure 5—figure supplement 3E*). Cells with homozygous disruptions of exon 2 and exon 8 showed no proliferation defect, no change in telomere length from the exon 2 KO, and no loss of Cajal bodies (*Figure 5—figure supplement 3F,G*). Overall, based on the assays described above, we conclude that TCAB1 and Cajal bodies are not essential for hTR telomerase to maintain telomeres at a stable length homeostasis.

## TCAB1 is not required for rapid telomere elongation by telomerase overexpression

In human somatic cells that remain telomerase-positive, stimulation to proliferate can be accompanied by dramatic telomerase activation and telomere length gain (*Weng et al., 1997*). This rapid increase in telomere length could depend on telomerase assistance by TCAB1 and/or Cajal bodies in a manner not required for maintaining length homeostasis. We therefore tested TCAB1 and Coilin KO cells for their ability to support rapid telomere elongation upon an increase in telomerase expression, achieved by integrating transgenes for hTR and TERT overexpression at AAVS1.

The parental, TCAB1 KO, and Coilin KO HCT116 cells with integrated hTR and TERT transgenes acquired ~2-fold elevated hTR and ~5-fold elevated telomerase catalytic activity (*Figure 6A*). These High-Telomerase (HiT) polyclonal cell cultures had increased telomere length by the first time point after selection (*Figure 6B*). Telomere elongation was rapid in TCAB1 KO HiT cell cultures, reaching the limit of length discrimination almost immediately. Rapid telomere elongation in TCAB1 KO cells is consistent with telomere elongation in wild-type HeLa cells by overexpression of CAB-box-mutant hTR (*Fu and Collins, 2007*; *Cristofari et al., 2007*). In comparison, Coilin KO HiT cells had less telomere elongation (*Figure 6B*). These results were replicated in an independently performed HiT conversion of the same cell lines (not shown). Telomeres in both TCAB1 KO and Coilin KO cells elongated upon expression of a truncated POT1 compromised for binding to single-stranded telomeric-repeat DNA (*Loayza and De Lange, 2003*), expressed by transgene integration at AAVS1 (data not shown). No obvious hTR foci were detected in HCT116 HiT cells lacking TCAB1 or Coilin (*Figure 6C*), indicating that the hTR foci detected in HeLa cells lacking Coilin (*Chen et al., 2015*) are challenging to detect. We conclude that if telomerase is abundant, TCAB1 is not required to support telomerase-mediated elongation of even relatively long telomeres. Integrating all of the findings of this study, we conclude that human telomerase H/ACA RNP assembly is essential not only for

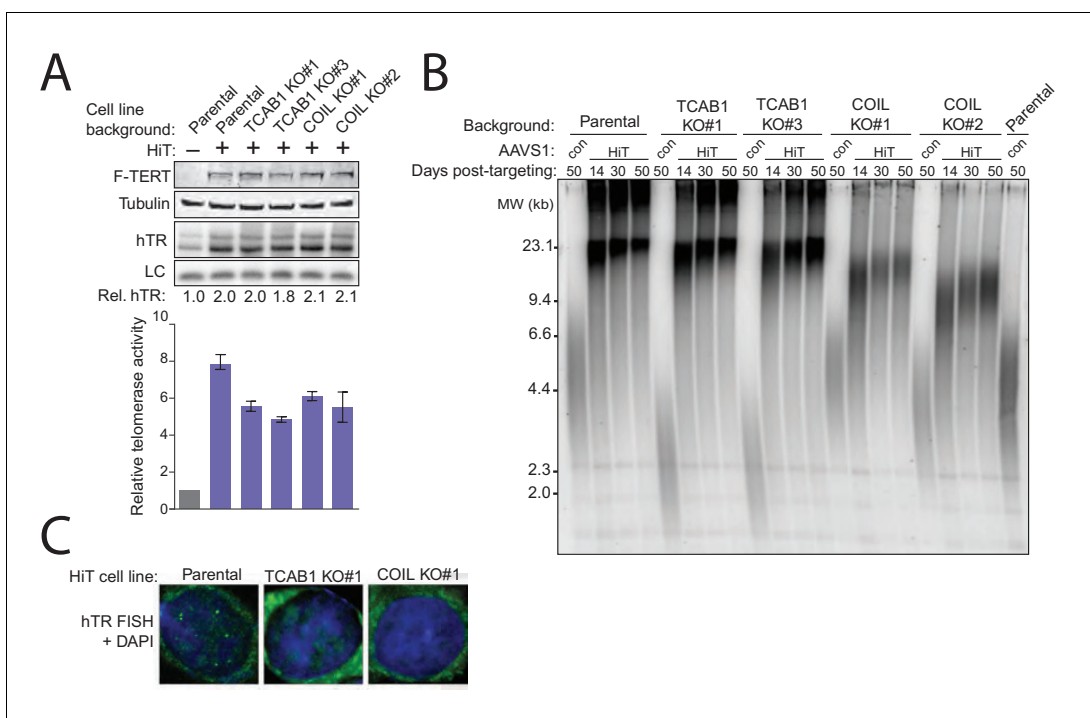

**Figure 6.** Cajal bodies promote telomere elongation upon increased telomerase expression level. (**A**) Immunoblot, northern blot, and QTRAP characterization of TCAB1 KO and COIL KO HCT116 cells overexpressing hTR and TERT at AAVS1 (HiT). QTRAP values were normalized to parental HCT116 (n = 3). (**B**) Time course of TRF in the HiT HCT116 TCAB1 KO and COIL KO cell cultures polyclonal following HiT transgene introduction. Days post-targeting refers to HiT transgene introduction. The lanes labeled 'con' are the indicated cell line without HiT transgene introduction. (**C**) FISH for hTR localization in HiT HCT116 cell lines.

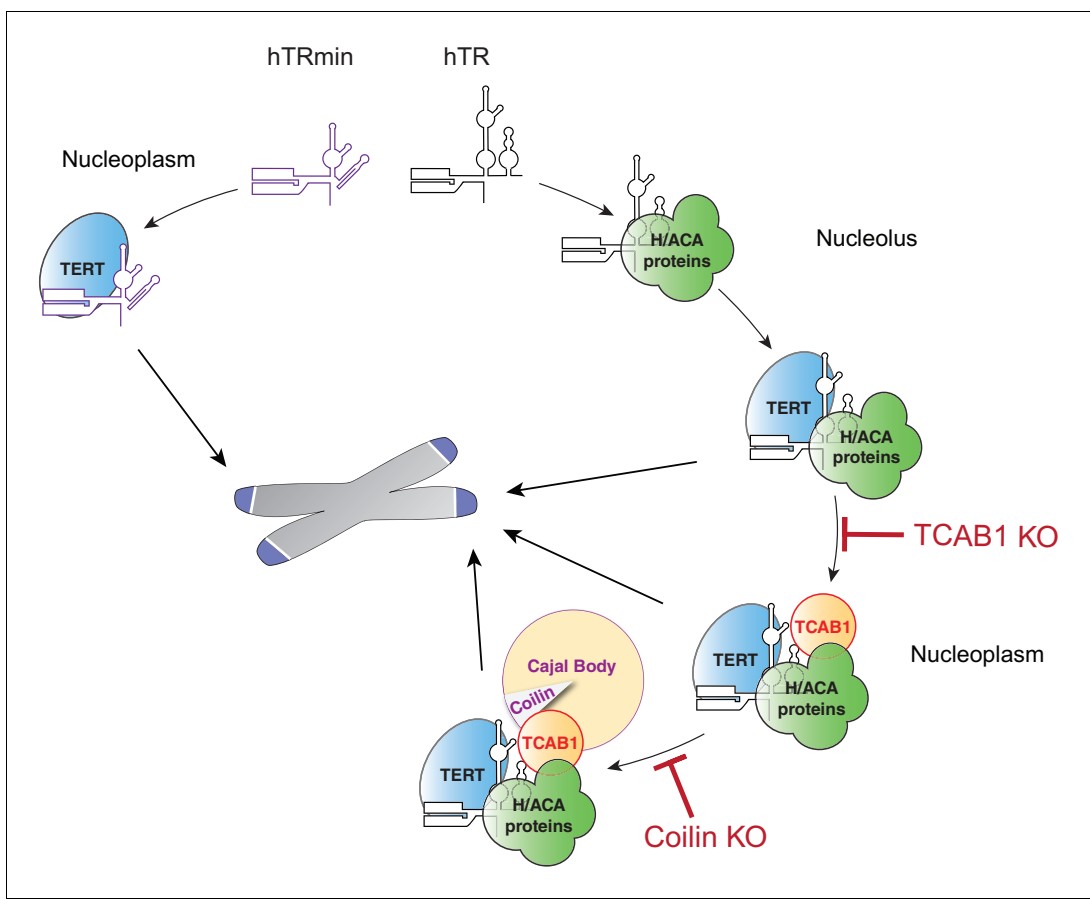

**Figure 7.** Multiple traffic pathways for human telomerase biogenesis and action at telomeres. At left, hTRmin telomerase assembles and acts at telomeres without supplemental trafficking instructions. At right, endogenous hTR and TERT are trafficked for their assembly and for telomerase action at telomeres.

the biological stability of hTR but also for active RNP biogenesis at scarce subunit expression levels. In addition, H/ACA RNP assembly gives endogenous-level hESC telomerase the ability to maintain long telomeres. Remarkably, all of these H/ACA RNP assembly roles can be bypassed using hTRmin (*Figure 7*). This plasticity of telomerase RNP biogenesis and holoenzyme composition informs mechanisms of telomerase diversification across eukaryotes.

## Discussion

Telomerase holoenzymes from different eukaryotes share the cellular requisites of stable RNP biogenesis and active RNP recruitment to telomeres in S-phase, but the mechanisms that underlie these requirements for telomerase function vary greatly (*Egan and Collins, 2012a*). We set out to uncover the rationale for a vertebrate telomerase evolutionary gain of H/ACA RNP biogenesis. H/ACA RNP biogenesis confers hTR biological stability, but across eukaryotes, telomerase RNA stability can be conferred by diverse other RNP assembly pathways. To bypass the essential role of H/ACA proteins in hTR cellular stability, we designed hTRmin RNAs containing the activity-essential hTR motifs and a 3' triplex structure but not an H/ACA motif. Although hTRmin did not bind H/ACA proteins or TCAB1, it did assemble catalytically active telomerase. The opposite TER redesign strategy was used to test the significance of the relative positioning of a large number of holoenzyme protein binding motifs in *S. cerevisiae* TLC1 (*Zappulla et al., 2005*). *S. cerevisiae* Mini-T retains all of the holoenzyme protein interactions but condenses the length of spacing between them (*Zappulla et al., 2005*).

Initially we expected the biological function of hTRmin-TERT RNPs to require tethering to Cajal bodies or telomeres. Instead, hTRmin-TERT RNPs supported stable telomere maintenance without any supplementary trafficking instructions (*Figure 7*, left). Therefore, most telomerase holoenzyme proteins may serve indirect roles in telomerase function that are readily adapted to evolutionary divergence of nuclear architecture and its cell cycle regulation. Other than hTR biological stability, the contributions of H/ACA RNP assembly, TCAB1, and Cajal bodies to endogenous human telomerase function could all be accounted for by changes in subunit and RNP distribution within the nucleus (*Figure 7*, right).

Studies here point to stabilization and efficient interaction of low-abundance telomerase subunits as a main rationale for the vertebrate telomerase H/ACA RNP biogenesis pathway. Cellular assembly of minimized hTRmin into catalytically active telomerase RNP required a higher than endogenous expression level of TERT. One simple model for this influence would be that H/ACA RNP biogenesis directs trafficking of assembly-competent hTR to meet assembly-competent TERT. Based on the cell cycle regulation of human telomerase subunit synthesis and subunit exchange in assembled RNPs, we have suggested that active RNP assembly may occur with only newly synthesized TERT and/or hTR subunits prior to their steady-state distribution (*Vogan and Collins, 2015*). Newly synthesized subunits could meet in Cajal bodies, as pre-mature hTR transits to acquire 5' cap trimethylation, or in nucleoli. Additional non-exclusive possibilities for cellular role(s) of hTR H/ACA RNP assembly in active RNP biogenesis include hTR folding, which would parallel the precedent from ciliate telomerase biogenesis (*Stone et al., 2007*), and/or exclusion of non-productive hTR-protein interactions.

In both HCT116 cells and hESCs, TCAB1 KO led to telomere shortening followed by telomere length homeostasis. As one model to account for these results, TCAB1 could influence the subnuclear distribution of active RNP between nucleoli and nucleoplasm (*Figure 7*, right). This change in distribution would have different influence on the amount of new telomeric repeat synthesis depending on telomerase expression level and other variables across human cell types. Of note, we have not ruled out an influence of TCAB1 related to its proposed function in DNA damage repair (*Henriksson and Farnebo, 2015*). However, in studies here it is striking that TCAB1 KO affected telomere length only in cells with hTR telomerase, not in cells with hTRmin telomerase. Therefore, TCAB1 functions directly related to active telomerase assembly and trafficking are more logical hypotheses.

Consistent with a contemporary study of Coilin KO in HeLa cells (*Chen et al., 2015*), we found that Coilin KO in HCT116 cells or in hESCs was permissive for stable telomere length maintenance. Coilin KO clonal cell lines generated in this work had telomere lengths similar to the parental cell lines. Because Coilin KO cells with endogenous telomerase levels have apparently unperturbed telomere length homeostasis, we suggest that they will be useful for visually tracking telomerase interactions with telomeres without the complication of Cajal body clustering of active with inactive hTR. Curiously, Coilin KO HCT116 cells had dampened telomere elongation upon an increase in telomerase expression. This could reflect less synergistic telomerase RNP loading on telomeres. Alternately, Cajal bodies could coordinate telomerase with other factors important for telomere synthesis, or they could safeguard telomere integrity. Our results support a paradigm of nuclear bodies as zones that draw factors together to fine-tune the likelihood of their physical association or functional cooperation rather than their interaction or reaction specificity.

Previous studies did not anticipate the genetic consequences of TCAB1 and Coilin KOs. For example, opposite the impact of TCAB1 and Coilin KOs on telomere lengths described above, previous studies found that endogenous hTR colocalization with telomeres was reduced more by Coilin depletion than TCAB1 depletion and overexpressed hTR colocalization with telomeres was reduced more by TCAB1 depletion than by Coilin depletion (*Zhong et al., 2012*; *Stern et al., 2012*). Also, TCAB1 mutations result in severe DC (*Zhong et al., 2011*; *Armanios and Blackburn, 2012*). We suggest that TCAB1 mutations reduce telomere length in early human development and in somatic cells with long telomeres. Overall, insights from this work inform strategies of therapy for human disease.

## Materials and methods

### Cell culture

HCT116, 293T, VA-13, and U2OS cells were cultured in DMEM with GlutaMAX (Thermo, Waltham, MA) supplemented with 10% FBS and 100 µg/ml Primocin (Invivogen, San Diego, CA). HCT116 cells were obtained from the Molecular and Cell Biology Department tissue culture facility (UC Berkeley). The 293T, VA-13, and U2OS cell lines are long-term Collins lab stocks. The hESC line WIBR#3 (NIH stem cell registry #0079, originating at the Whitehead Institute for Biomedical Research) was maintained on inactivated mouse embryonic fibroblasts in DMEM/F12 under conditions previously described (*Chiba and Hockemeyer, 2015*). None of the cell lines have been authenticated recently at the genome sequence level, but all have the expected cell morphology and doubling time. HCT116 and hESC cell lines were tested for mycoplasma. Transient transfection, was performed using calcium phosphate or Lipofectamine 2000 (Thermo).

### Northern blots

RNA was purified using TRIzol (Thermo) and resuspended in distilled water. Concentration was determined by absorbance at 260 nm. Formamide loading buffer was added to RNA samples, followed by heat denaturation at 95°C for 1 min and ice for 5 min before loading on a denaturing gel (5% of 19:1 acrylamide:bis-acrylamide, 0.6X TBE, 32% formamide, and 5.6 M urea). Following electrophoresis, RNA was transferred to nylon membrane by electroblotting. RNA was then UV crosslinked and the membrane was blocked in Church's buffer (1% BSA, 1 mM EDTA, 0.5 M $NaPO_4$, 7% SDS) with 15% formamide at 50°C. $^{32}$P-labeled, 2'O-methyl RNA probe complementary to the hTR template region was then added (*Fu and Collins, 2003*). In some experiments, $^{32}$P-labeled DNA oligonucleotides complementary to regions in the hTR pseudoknot (5'-TAGAATGAACGG TGGAAGGCGGCAGGCCGAGGCT-3') and hTR CR4/5 (5'-TCGCGGTGGCAGTGGGTGCCTCCGGA-GAAGCCC-3') were also added to enhance signal detection. Membranes were hybridized overnight at 50°C, followed by extensive washing with 1X SSC + 0.1% SDS at 50°C. Blots were then exposed on phosphorimager screens and subsequently scanned on a Typhoon (GE Healthcare, Chicago, IL). Scans were processed using ImageJ software (NIH). The Northern blot loading control (LC) is a non-specifically detected cellular RNA.

### Immunoblots

Protein was resolved in 10% Bis-Tris SDS-PAGE gels in MOPS buffer (250 mM MOPS pH 7.0, 250 mM Tris, 5 mM EDTA, 0.5% SDS) before being transferred to nitrocellulose membrane. The membranes were then blocked in immunoblot buffer consisting of 4% nonfat milk (Carnation) in TBS buffer (150 mM NaCl, 50 mM Tris pH 7.5). Membranes were then incubated with primary antibodies diluted in immunoblot buffer at 4°C overnight. After primary antibody incubation, membranes were washed extensively in TBS, followed by secondary antibody incubation in immunoblot buffer for 1 hr at room temperature. After washing in TBS, blots were scanned using a LI-COR Odyssey imager. Immunoblot primary antibodies included mouse anti-tubulin (1:500, DM1A, Calbiochem, Billerica, MA), mouse anti-FLAG (1:4,000, F1804, Sigma), rabbit anti-TCAB1 (1:2000, NB100-68252, Novus Biologicals, Littleton, CO), mouse anti-Coilin (1:250, IH10, Abcam, Cambridge, United Kingdom), and mouse anti-TERT (1:2000 (*Wu et al., 2015*)). Secondary antibodies included goat anti-mouse Alexa Fluor 680 (1:2000, Life Technologies, Carlsbad, CA) and goat anti-rabbit IR Dye 800 (1:10,000, Rockland Immunochemicals, Pottstown, PA).

### Immunopurification

HLB150 buffer (20 mM HEPES pH 8.0, 2 mM $MgCl_2$, 0.2 mM EGTA, 10% glycerol, 1 mM DTT, 0.1% Igepal, Sigma protease inhibitor cocktail, 150 mM NaCl) was used throughout. Samples were normalized to 2 mg/ml and 200 µl was incubated at 4°C for 1–2 hr with 4 µl magnetic anti-FLAG M2 beads (Sigma) or 4 µl magnetic Protein A/G beads (BioTool, Houston, TX) pre-bound with antibody against TCAB1 (Novus Biologicals), antibody against FLAG (Sigma), or rabbit IgG (Sigma). After binding and washing, the beads were resuspended in 20 µl HLB150 buffer. For hotTRAP, the resuspended beads were diluted 1:10 in HLB150, and 2 µl of the diluted sample was used per hotTRAP reaction. For Northern blot analysis, the washed beads were processed with TRIzol (Thermo).

## Microscopy

Live cell imaging was performed on a Zeiss Axio Observer A1 with a 40X phase contrast objective. For RNA and protein localization, cells were fixed in 4% paraformaldehyde for 10 min at room temperature. Cells were then washed with PBS twice, before additional fixation and permeabilization with 100% methanol for 10 min at room temperature. For RNA FISH, the methanol was aspirated and RNA probes in RNA FISH buffer were directly added to cells. Detection of hTR and hTRmin used Cy3-labeled RNA probes complementary to the template, pseudoknot, and CR4/5 spanning hTR nt 36–70, 129–162, and 249–281, respectively. Ten ng of each probe was diluted in RNA FISH buffer consisting of 2X SSC, 10% formamide, and 10% dextran sulfate. Probes were incubated with samples overnight at 37°C in a humidified chamber. Samples were then washed 6 times with 2X SSC before being mounted on coverslips using ProLong Gold with DAPI (Thermo). For immunofluorescence, cells were washed three times in PBS following methanol incubation and then rehydrated and blocked in 4% BSA in PBS for 1 hr at room temperature or overnight at 4°C. Primary antibodies diluted in 4% BSA in PBS were then added and incubated with samples for 1 hr at room temperature. Primary antibodies used included rabbit anti-TCAB1 (1:300, NB100-68252, Novus Biologicals), mouse anti-Coilin (1:250, IH10, Abcam), and mouse anti-SMN (1:300, sc-15320, Santa Cruz Biotechnology, Santa Cruz, CA). After washing in PBS, samples were incubated with Alexa Fluor 488 or Alexa Fluor 568 (Thermo) secondary antibodies diluted in 4% BSA in PBS for 1 hr at room temperature. Samples were then washed in PBS and mounted on coverslips using ProLong Gold with DAPI. Images were acquired using a Zeiss LSM510 Meta confocal microscope with a ×100/1.49 Apo objective, with 364-, 488-, and/or 543-nm laser excitation.

## Telomerase activity assays

Cell extract was prepared by hypotonic freeze thaw lysis and salt extraction. Primer extension and hotTRAP were performed as previously described (*Sexton et al., 2014*). Unless otherwise noted, 200 ng total protein was used per hotTRAP or QTRAP reaction, quantified using the Bio-Rad protein assay. An internal control (IC) to normalize PCR amplification was always included in hotTRAP assays. QTRAP used the iTaq universal SYBR green Supermix (Bio-Rad) and a CFX96 Touch Real-Time PCR Detection System (Bio-Rad, Hercules, CA) as previously described (*Vogan and Collins, 2015*). Relative telomerase activity measured by QTRAP was calculated by delta Ct to a reference sample. All error bars shown are standard error of the mean and statistical significance was calculated using ANOVA with Tukey's multiple comparison test in GraphPad Prism 6.

## Genome engineering

Single-guide RNA sequences for Cas9 targeting of protein coding genes were designed for selection cassette insertion downstream of the start codon. Optimal guide sequences were generated using the CRISPR Design tool (http://crispr.mit.edu/). Guide sequences were then inserted into the PX330 plasmid (*Cong et al., 2013*). TERC guide sequence, followed by the PAM: 5'-TTCAGCGGGCG-GAAAAGCCT CGG-3'. TCAB1 exon 2 guide sequence, followed by the PAM: 5'-TTTATTCA TCGGGGAAGCGT GGG-3'. TCAB1 exon 8 guide sequence, followed by the PAM: 5'-TGAGAA-GAAGCGGTTGCCAT CGG-3'. COIL exon 1 guide sequence, followed by the PAM: 5'-AAGCCG TAGCCTAACCGTCT CGG-3'. Donor plasmid constructs were designed by flanking a puromycin resistance cassette or a hygromycin resistance cassette with sequence 500–600 bp upstream and downstream of the genomic DNA cut site. The TCAB1 targeting sites do not overlap with the upstream p53 gene.

Zinc-finger nuclease (ZFN) mediated disruption of the TERT gene and transgene integration at AAVS1 have been previously described (*Sexton et al., 2014*). For the disruption of TERT gene exon 1, a donor plasmid carrying a hygromycin resistance cassette flanked by homology arms of approximately 500 bp upstream and downstream of the cut site was transfected with plasmids expressing the TERT-targeting ZFN. For AAVS1 transgene integration, donor plasmids carrying the transgene(s) with an upstream neomycin or puromycin resistance cassette, together flanked AAVS1 homology arms, were transfected with plasmids expressing the AAVS1-targeting ZFN.

Genome engineering in HCT116 cells was performed by Lipofectamine 3000 transfection according to manufacturer guidelines (Thermo) using a 2:1 ratio of donor plasmid to nuclease plasmid. HCT116 cell lines were selected with 300 μg/ml hygromycin, 1 μg/ml puromycin, or 300 μg/ml

G418. hESC gene editing was performed by electroporation as previously described (*Sexton et al., 2014*; *Chiba and Hockemeyer, 2015*).

## Lentiviral transduction

Lentivirus was produced in 293T cells by calcium phosphate transfection with the packaging plasmid, psPAX2, the envelope plasmid, pMD2.G, and with transgene constructs in the DUET011 backbone (*Zhou et al., 2007*). Cell transfection media was replaced at 24 hr post-transfection and virus was harvested 48 hr post-transfection. Virus-containing media was applied to HCT116 cells in the presence of 5 µg/ml polybrene (Sigma) for 24 hr before a media change. At 48 hr post-infection, transduced cells were selected with 300 µg/ml hygromycin.

## PCR genotyping

Genomic DNA was prepared as described above for telomere restriction fragment analysis. Between 50–100 ng of genomic DNA was used as the template for PCR using the Q5 polymerase (NEB, Ipswich, MA). For PCR confirmation of gene editing, PCR primers were designed to generate amplicon size differences between loci with or without a drug resistance cassette. Paired PCR primers complementary to genomic loci had one primer complementary to a region also in the donor plasmid and the other primer complementary to a region beyond the donor plasmid homology. A third primer against either the hygromycin or puromycin resistance cassette was included that would generate an amplicon size for cassette-containing alleles that was either smaller or larger than amplicons from alleles lacking the cassette.

For TCAB1 exon 2 and Coilin exon 1 PCRs, a hygromycin cassette primer (PGK hygro: 5'-AGGC TGATCAGCGGTTTAAACTTAGCCTCCCCTACCCGGTAGAATTC-3'; or hygro pA: 5'-CTAGTGGA TCCGAGCTCGGTACCAGATGCGGTGGGCTCTATGGC-3') was paired with the following locus-specific primers:

Coil_FWD: 5'-TAGTGGATCCGAGCTCGGTACCACCACTGCTCCTGGCCTCTAGTTAC-3', Coil_-REV: 5'-AGGCTGATCAGCGGTTTAAACTTAAGAACTGAAGCCGAAGCGCTGG-3', TCAB_FWD1: 5'-CTAGTGGATCCGAGCTCGGTACCAGGAAGGCTTTCCGTAATATCACACCCTAACG-3', and TCAB_REV1: 5'-AGGCTGATCAGCGGTTTAAACTTACAGAAAGTTCTTGCTCCTCGATTCGAGGAC TC-3'. An alternative set of TCAB1 locus primers was also used for additional validation of the lines: TCAB1_FWD2: 5'-CTAGTGGATCCGAGCTCGGTACCAGCGGTGCTAAGGAACACAGTGC TTTCAAAAG-3', and TCAB1_REV2: 5'-AGGCTGATCAGCGGTTTAAACTTAGGCATCCCTCTCC TAGAAAACTGG-3'.

For TCAB1 exon 8 PCR, a primer against the puromycin resistance cassette (PGK_PURO: 5'-GGCGCACCGTGGGCTTGTACTCGGTCATGGTGGCGGGATGCAGGT CGAAAGGCCCG-3') was combined with two loci primers (TCAB1_ex8_FWD: 5'-CCAAGGCCAACCAGCTGGTCAAAGGAC TGCTTC-3', and TCAB1_ex8_REV:5'-CTCAGCATCCTGGAGACAAGGAACAGGACCTGGAGT-3'). For TERT locus PCR, another primer against the hygromycin cassette was used (TERT_hygro: 5'-C TCACCGCGACGTCTGTCGAGAAG-3') with the following locus-specific primers: TERT_FWD 5'-C TAGTGGATCCGAGCTCGGTACCAGCGGCGCGAGTTTCAGGCAG-3', and TERT_REV 5'-AGGC TGATCAGCGGTTTAAACTTAAACGGCAGACTTCGGCTGGCAC-3'.

For TERC locus PCR, a puromycin resistance cassette primer (5'-TGAAGCCGAGCCGCTCG TAGAA-3') was combined with locus-specific primers: hTR_FWD 5'-GTGGATCCGAGCTCGGTAC-CACCCACTGAGCCGAGACAAGATTC-3' and hTR_REV 5'-GAAAGCGAACTGCATGTGTGAGCCG-3'. Some primers had 5' regions complementary to the pcDNA3.1 vector to facilitate cloning, for which DNA was gel-excised, purified, and ligated into pcDNA3.1 (Thermo). DH5a cells were then transformed and several colonies were sequenced.

## Southern blots

Cell pellets were lysed in RIPA buffer (150 mM NaCl, 50 mM Tris at pH 7.5, 1 mM EDTA, 1% Triton X-100, 0.5% sodium deoxycholate, 0.1% SDS, 1 mM DTT), treated with RNase A for 30 min at 37°C, followed by proteinase K treatment for 4 hr to overnight at 50°C. DNA was then purified using phenol-isoamyl alcohol-chloroform, followed by precipitation with isopropanol and NaCl. After pelleting, the DNA was washed twice with ethanol and resuspended in TE buffer (10 mM Tris, 1 mM EDTA). DNA concentration was determined by absorbance at 260 nm. Two to 8 µg of DNA was then

digested with MboI and AluI for 6 hr to overnight before electrophoresis in a 0.7% agarose gel in 1X TAE. The agarose gel was then vacuum dried at 50°C for 1 hr. The dried gel was then denatured with 0.5 N NaOH, 1.5 M NaCl for 30 min at 50°C. The gel was washed twice with 4X SSC + 0.1% SDS and subsequently blocked with Church's buffer for 30 min at 50°C. $^{32}$P-end-labeled telomeric repeat probe (T$_2$AG$_{3)3}$ and $^{32}$P-labeled probe made by random-priming of HinDIII lambda phage digest ladder (NEB) and/or the O'generuler 1 kbp plus DNA ladder (Thermo) were added. Probes were hybridized overnight at 50°C. The membrane was then extensively washed in 4X SSC + 0.1% SDS at 40°C before screen exposure and imaging on a Typhoon scanner (GE Healthcare).

## Acknowledgements

We thank Jessica Brown and Joan Steitz for the M1 expression vector. We thank Jane Tam for help with cell culture.

## Additional information

### Competing interests

KC: Reviewing editor, *eLife*. The other authors declare that no competing interests exist.

### Funding

| Funder | Grant reference number | Author |
| --- | --- | --- |
| National Heart, Lung, and Blood Institute | RO1 HL079585 | Kathleen Collins<br>Jacob M Vogan<br>Xiaozhu Zhang<br>Daniel T Youmans |
| Ellison Medical Foundation | | Dirk Hockemeyer |
| Glenn Foundation for Medical Research | | Dirk Hockemeyer |

The funders had no role in study design, data collection and interpretation, or the decision to submit the work for publication.

### Author contributions

JMV, Performed all experiments other than hESC gene knockouts and cell cultures, Conducted experimental design and analysis, Wrote the manuscript; XZ, Performed all experiments other than hESC gene knockouts and cell cultures; DTY, Developed the initial strategy for hTRmin expression in cells; SGR, Generated and characterized the hESC lines; JZJ, Developed the initial strategy for TERC locus disruption; DH, Conducted experimental design and analysis, Revised the manuscript; KC, Conducted experimental design and analysis, Wrote the manuscript

### Author ORCIDs

Kathleen Collins, http://orcid.org/0000-0003-3172-7088

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
