## [Decision Letter]

Thank you for submitting your article "Streamlined human telomerases maintain telomeres and reveal endogenous roles of H/ACA proteins, TCAB1, and Cajal bodies" for consideration by eLife. Your article has been favorably evaluated by James Manley (Senior editor) and three reviewers, one of whom, Carol Greider (Reviewer #1), is a member of our Board of Reviewing Editors. The following individual involved in review of your submission has agreed to reveal their identity: Jonathan Alder (Reviewer #2).

The reviewers have discussed the reviews with one another and the Reviewing Editor has drafted this decision to help you prepare a revised submission. We hope you will be able to submit the revised version within two months, so please let us know if you have any questions first.

This manuscript by Vogan et al. provides strong evidence that a minimal telomerase RNA functions to elongate telomere in vivo in human cells. Recognizing that over evolution, telomerase RNAs in different species have acquired different 3' ends, the author test whether the box H/ACA 3' region of hTR is needed for telomerase to function at telomeres. They demonstrate that the H/ACA-interacting regions of hTR are dispensable if modifications are made to stabilize the RNA. The authors use the cleaver trick of putting an alternative stable 3' end from the MALAT1 triplex structure on the 3' end of hTR and find that when expression levels are sufficiently high, this minimal hTR functions to elongate telomeres in transformed human cells and hES cells. They go on to test the importance of TCAB1 and Coilin in telomere length maintenance and show they are not required for telomere elongation in vivo. This paper is important because it establishes that the minimal region for the human telomerase RNA does not require the 3' region or several accessory proteins. The work expands our understanding of the roles of TCAB1 and Coilin, which have been the subject of some controversy.

The manuscript does not need additional experiments but requires additional clarification of experiments presented and should be significantly rewritten to make it clearer and more concise. In places the manuscript is a bit confusing to follow and there are a number of changes that will greatly help clarify the conclusions.

Specific scientific points:

1) In the Introduction the authors should site Zappulla 2005 Nat Struct Mol Biol. 2005. This group showed that a minimal telomerase RNA also functions in yeast, and similar to what is shown here, this smaller RNA needs to be expressed at a higher level perhaps to overcome inefficient assembly. Referring to this previous work in yeast would strengthen this argument made in this manuscript as the minimal RNA requirement is conserved.

2) In the second paragraph of the subsection “Human telomerase can be liberated from its H/ACA RNP identity”: the description of the different version of the hTR min here is very confusing. This section should be rewritten to clarify what each construct has, and not just say 'we tested various lengths..." Lay out each component (RNA domain) and why it is being used, and explain what the "LeaderG means. Figure 1 shows this well, but need to write about this clearly.

3) It is not clear from Figure 2 whether telomerase activity can be increased with increasing amounts of hTRmin expression (as it can be with hTR Figure 1—figure supplement 1), has this been tested? Or is it only hTERT over expression that stabilizes hTRmin?

4) In the fourth paragraph of the subsection “Catalytically active telomerase supports telomere maintenance without H/ACA proteins or TCAB1” and Figure 2: The authors state that the hTR KO cell lines lacking active telomerase ultimately entered senescence, however no time frame or passage number is given. Ideally clonogenic survival or growth curve would be appropriate to present. Simple pictures of cells cannot demonstrate the heterogeneity in the population of cells thus Figure 2 is not needed. Showing this picture does not help the reader, better to know how long it took the cells to die. Some discussion of this heterogeneity in viability would be very helpful. Also it would be very helpful to state passage numbers or days in culture to the 'time' arrow at the bottom of 2F.

5) Specifically it is important to understand whether some cells in the population continue to grow for some time? Are some cells able to escape short telomere phenotype cause by hTRmin or LhTRmin?

6) In the fifth paragraph of the subsection “Catalytically active telomerase supports telomere maintenance without H/ACA proteins or TCAB1”: The authors state, "curiously these clonal lines had stable but distinct set points". I am not sure why they say 'curiously' as it is well established in the literature that the founder effect of a given telomere length sets the length distribution in the daughter cells and then the equilibrium is distributed around that founding length. This fact has been published by others in human cells and in fact was first seen in yeast (Shampay & Blackburn PNAS 1988 Jan;85(2):534-8. Generation of telomere-length heterogeneity in *Saccharomyces cerevisiae*).

7) In Figure 3—figure supplement 1 the telomere lengths in cells expressing LhTRmin increase at 79 days post-targeting. How do these kinetics of elongation compare with time of elongation for expression of hTR (only 1 time point shown in Figure 3 for hTR)? Could the level of RNA expression affect the rates of elongation?

8) In Figure 3 can the authors comment on the possible sources of variable expression of F-TERT. Presumably, these constructs have been integrated into the AAVS1 safe-harbor site. Do the differences arrive from different levels of the various hTRs?

9) In the third paragraph of the subsection “TCAB1 and Cajal bodies contribute distinct, non-essential regulations of telomerase”: As noted above, heterogeneity in set points does not necessarily have to come from variation of telomerase component levels. The founder effect can establish differences in telomere length distributions without altering the genetically determined set point equilibrium. This concept of concluding this is an altered 'set point' should be reconsidered throughout the manuscript. Their data for hTRmin function is strong, there is no need to confuse it with discussion of alterations of set point.

10) In the fifth paragraph of the subsection 2TCAB1 and Cajal bodies contribute distinct, non-essential regulations of telomerase”: The conclusion that TCAB1 allows cells with box H/ACA to maintain longer telomere is presented without stating what the RNA levels are. What are the hTR levels and the stability? Is there a shorter half-life of hTRmin? I do not think additional experiments are needed, however more concise conclusions would help. The longer telomeres may be due to a more stable RNA, not any function of TCAB1. This sentence/conclusion should be reworked.

11) In the last paragraph of the subsection “TCAB1 and Cajal bodies contribute distinct, non-essential regulations of telomerase”: the conclusion that is made "that TCAB and Cajal bodies are not needed to maintain telomeres at a stable length" does not accurately reflect the data shown. In the figure, clearly, telomeres get longer (Figure 5—figure supplement 2). The authors should say something about this elongation and then make conclusion from that, before saying perhaps that minimal maintenance of telomeres does not require TCAB and Cajal bodies. But to ignore the large increase in length in the concluding sentence is dissonant with the results presented.

12) In the second paragraph of the subsection “TCAB1 and Coilin differentially affect telomerase ability to increase telomere length”: the authors state there is "a 5 fold elevated telomerase catalytic activity".

However, TRAP is measured so they should state an increase in TRAP products, as they are not measuring telomerase catalytic activity rate.

13) In the second paragraph of the subsection “TCAB1 and Coilin differentially affect telomerase ability to increase telomere length”: The authors state, Coilin KO HiT cells exhibited much slower and dramatically blunted telomere elongation" this statement does not reflect the data in Figure 6. While there is some difference in telomere elongation rate in TCAB1 KO and Colin KO cells, the overwhelming effect that is seen in these gels is dramatic telomere elongation. Some statement about the significant elongation is required for the reader to understand what subtle difference the authors are pointing to between the slightly different extents of the significant elongation. The words 'dramatically blunted' do not fit the data that is shown.

14) In the second paragraph of the subsection “TCAB1 and Coilin differentially affect telomerase ability to increase telomere length”: the authors say "this is the first time a factor other than telomerase determine the extent of length gain". This sentence should be removed. It is well known that altering shelterin component levels affects length equilibrium, and it has been shown by several groups that ATM kinase activity regulates telomere elongation.

15) In the last paragraph of the subsection 2TCAB1 and Coilin differentially affect telomerase ability to increase telomere length”: The authors conclude from Figure 6 that there is no effect of Coilin KO on the level of telomere elongation in a Pot1 DeltaOB mutant background. However, it is striking that there is a strong effect in the TCAB KO that the authors do not comment on. Comparing Figure 6 and Figure 6, it seems there is an equal but opposite effect of TCAB and Coilin KO. There is less elongation in Figure 6 in the Coilin mutant, and in Figure 6 there is less elongation in the TCAB mutant. This is very confusing and the authors do not mention this at all. I would suggest that since the Pot1 DeltaOB mutant is not really a large part of this story, that this experiment should be removed from the manuscript. It does not add any important information and is not easy to interpret.

---

## [Author Response]

The manuscript does not need additional experiments but requires additional clarification of experiments presented and should be significantly rewritten to make it clearer and more concise. In places the manuscript is a bit confusing to follow and there are a number of changes that will greatly help clarify the conclusions.

Specific scientific points:

1) In the Introduction the authors should site Zappulla 2005 Nat Struct Mol Biol. 2005. This group showed that a minimal telomerase RNA also functions in yeast, and similar to what is shown here, this smaller RNA needs to be expressed at a higher level perhaps to overcome inefficient assembly. Referring to this previous work in yeast would strengthen this argument made in this manuscript as the minimal RNA requirement is conserved.

The Mini-T of Zappulla/Cech tests a different question, which is whether a yeast telomerase RNA has a required spacing of the holoenzyme protein components from each other. Their design strategy is the exact opposite of ours: Mini-T retains all of the holoenzyme protein interaction motifs condensed together, whereas hTRmin removes all motifs for holoenzyme protein interactions other than TERT and adds spacer to replace them. We appreciate the suggestion for this comparison, which we have added to the Discussion.

2) In the second paragraph of the subsection “Human telomerase can be liberated from its H/ACA RNP identity”: the description of the different version of the hTR min here is very confusing. This section should be rewritten to clarify what each construct has, and not just say 'we tested various lengths..." Lay out each component (RNA domain) and why it is being used, and explain what the "LeaderG means. Figure 1 shows this well, but need to write about this clearly.

We expanded the description in the text.

3) It is not clear from Figure 2 whether telomerase activity can be increased with increasing amounts of hTRmin expression (as it can be with hTR Figure 1—figure supplement 1), has this been tested? Or is it only hTERT over expression that stabilizes hTRmin?

TERT overexpression does not influence hTRmin accumulation (Northern blot in Figure 2, lanes 4-5 versus 7-8), which we now state in the text. We were not able to experimentally "tune up" the level of hTRmin expression in cells, because even its optimal expression context gives just enough RNA to readily detect. We added better labeling to Figure 1—figure supplement 1 to make this comparison more obvious.

4) In the fourth paragraph of the subsection “Catalytically active telomerase supports telomere maintenance without H/ACA proteins or TCAB1” and Figure 2: The authors state that the hTR KO cell lines lacking active telomerase ultimately entered senescence, however no time frame or passage number is given. Ideally clonogenic survival or growth curve would be appropriate to present. Simple pictures of cells cannot demonstrate the heterogeneity in the population of cells thus Figure 2 is not needed. Showing this picture does not help the reader, better to know how long it took the cells to die. Some discussion of this heterogeneity in viability would be very helpful. Also it would be very helpful to state passage numbers or days in culture to the "time" arrow at the bottom of 2F.

hTR KO cells lacking active telomerase underwent cell death, not senescence. Figure 2 shows the membrane blebbing phenotype of cell death. The text now details how long it took the hTR KO cells to die, both in days and population doublings, as well as retaining that information in the lane labeling of the Figure panels. This level of description was also added for the TERT KO cells, and others.

5) Specifically it is important to understand whether some cells in the population continue to grow for some time? Are some cells able to escape short telomere phenotype cause by hTRmin or LhTRmin?

No cells escape hTR KO or TERT KO without rescue of telomerase catalytic activity, as stated in the text. As a polyclonal population, cells expressing LhTRmin and TERT did either escape short telomeres OR the short telomere cells were outgrown by long telomere cells, which is the point of Figure 3—figure supplement 1. It takes ~100 days to see that change in culture population, whereas telomere lengthening by hTR rescue is almost immediate (added Figure 3—figure supplement 1). Once clonal cell lines are taken from the polyclonal population, telomere length remains relatively constant (Figure 3).

6) In the fifth paragraph of the subsection “Catalytically active telomerase supports telomere maintenance without H/ACA proteins or TCAB1”: The authors state, "curiously these clonal lines had stable but distinct set points". I am not sure why they say "curiously" as it is well established in the literature that the founder effect of a given telomere length sets the length distribution in the daughter cells and then the equilibrium is distributed around that founding length. This fact has been published by others in human cells and in fact was first seen in yeast (Shampay & Blackburn PNAS 1988 Jan;85(2):534-8. Generation of telomere-length heterogeneity in Saccharomyces cerevisiae).

We deleted the word “curiously.” The difference here versus previous studies is that clonal cell lines are isolated from a pool of cells each with its own independent genome editing event. Genome editing leaves an epigenetic scar that could be variable depending on the length of endogenous locus involved in homologous recombination.

7) In Figure 3—figure supplement 1 the telomere lengths in cells expressing LhTRmin increase at 79 days post-targeting. How dothese kinetics of elongation compare with time of elongation for expression of hTR (only 1 time point shown in Figure 3 for hTR)? Could the level of RNA expression affect the rates of elongation?

See explanation for point 5 above. We added kinetics of telomere elongation in cells expressing hTR as Figure 3—figure supplement 1 (telomeres are already as long as in the parental cells at 24 days post-targeting).

8) In Figure 3 can the authors comment on the possible sources of variable expression of F-TERT. Presumably, these constructs have been integrated into the AAVS1 safe-harbor site. Do the differences arrive from different levels of the various hTRs?

Figure 3 shows hTR levels, TERT levels, and active RNP levels quantified for each clonal cell line. Across independent genome editing events in different cells, transgenes integrated at AAVS1 acquire different TERT and hTR expression levels, not obviously correlated to each other. This suggests that cell-to-cell differences should be considered in this type of study (studies should report on more than a single clonal cell line). We do not understand the basis of the variation yet.

9) In the third paragraph of the subsection “TCAB1 and Cajal bodies contribute distinct, non-essential regulations of telomerase”: As noted above, heterogeneity in set points does not necessarily have to come from variation of telomerase component levels. The founder effect can establish differences in telomere length distributions without altering the genetically determined set point equilibrium. This concept of concluding this is an altered "set point" should be reconsidered throughout the manuscript. Their data for hTRmin function is strong, there is no need to confuse it with discussion of alterations of set point.

See answers to points 5 and 8 above. We replaced “set-point” with “at stable telomere length homeostasis.”

10) In the fifth paragraph of the subsection 2TCAB1 and Cajal bodies contribute distinct, non-essential regulations of telomerase”: The conclusion that TCAB1 allows cells with box H/ACA to maintain longer telomere is presented without stating what the RNA levels are. What are the hTR levels and the stability? Is there a shorter half-life of hTRmin? I do not think additional experiments are needed, however more concise conclusions would help. The longer telomeres may be due to a more stable RNA, not any function of TCAB1. This sentence/conclusion should be reworked.

We show that TCAB1 KO doesn’t affect telomerase activity level, so we had not included the Northern blots for hTR because it seemed redundant. We did hTR/hTRmin Northern blots for all of the KO backgrounds, and we see no influence of TCAB1 or Coilin KO on hTR/hTRmin level. We added Figure 5 to show a Northern blot for hTR level in TCAB1 and Coilin KO cells. We also expanded the comparison at the level of catalytic activity by showing hotTRAP cell extract titrations for HCT116 and hESC TCAB1 and Coilin KO cell lines (Figure 5—figure supplement 1 and Figure 5—figure supplement 2; we split out the HCT116 and hESC cell experiments to be Figure 5—figure supplement 1 and Figure 5—figure supplement 2).

11) In the last paragraph of the subsection “TCAB1 and Cajal bodies contribute distinct, non-essential regulations of telomerase”: the conclusion that is made "that TCAB and Cajal bodies are not needed to maintain telomeres at a stable length" does not accurately reflect the data shown. In the figure, clearly, telomeres get longer (Figure 5—figure supplement 2). The authors should say something about this elongation and then make conclusion from that, before saying perhaps that minimal maintenance of telomeres does not require TCAB and Cajal bodies. But to ignore the large increase in length in the concluding sentence is dissonant with the results presented.

Figure 5—figure supplement 2 (now 3) shows HCT116 and hESC cells with TCAB1 KO rescued by re-expression of TCAB1. These cells should have telomere elongation, if the KO was responsible for the telomere shortening.

12) In the second paragraph of the subsection “TCAB1 and Coilin differentially affect telomerase ability to increase telomere length”: the authors state there is "a 5 fold elevated telomerase catalytic activity".

However, TRAP is measured so they should state an increase in TRAP products, as they are not measuring telomerase catalytic activity rate.

It is standard to use TRAP assays as a measure for amount of telomerase activity. It does not imply rate. We have verified a 5-fold difference by both QTRAP and hotTRAP, but the exact amount of increase isn’t particularly significant in this experimental context

13) In the second paragraph of the subsection “TCAB1 and Coilin differentially affect telomerase ability to increase telomere length”: The authors state, Coilin KO HiT cells exhibited much slower and dramatically blunted telomere elongation" this statement does not reflect the data in Figure 6. While there is some difference in telomere elongation rate in TCAB1 KO and Colin KO cells, the overwhelming effect that is seen in these gels is dramatic telomere elongation. Some statement about the significant elongation is required for the reader to understand what subtle difference the authors are pointing to between the slightly different extents of the significant elongation. The words 'dramatically blunted' do not fit the data that is shown.

We removes “dramatically,” although in HiT transgene assays of many many different genotype backgrounds we have never seen telomeres in HiT cells stay below 20 kbp vs longer than can be resolved by the gel.

14) In the second paragraph of the subsection “TCAB1 and Coilin differentially affect telomerase ability to increase telomere length”: the authors say "this is the first time a factor other than telomerase determine the extent of length gain". This sentence should be removed. It is well known that altering shelterin component levels affects length equilibrium, and it has been shown by several groups that ATM kinase activity regulates telomere elongation.

Sentence deleted. See answer to 13 for the point we were trying to make, which is not necessary to the story.

15) In the last paragraph of the subsection 2TCAB1 and Coilin differentially affect telomerase ability to increase telomere length”: The authors conclude from Figure 6 that there is no effect of Coilin KO on the level of telomere elongation in a Pot1 DeltaOB mutant background. However, it is striking that there is a strong effect in the TCAB KO that the authors do not comment on. Comparing Figure 6 and Figure 6, it seems there is an equal but opposite effect of TCAB and Coilin KO. There is less elongation in Figure 6 in the Coilin mutant, and in Figure 6 there is less elongation in the TCAB mutant. This is very confusing and the authors do not mention this at all. I would suggest that since the Pot1 DeltaOB mutant is not really a large part of this story, that this experiment should be removed from the manuscript. It does not add any important information and is not easy to interpret.

We deleted the POT1 dOB experiment. We mention that telomeres elongate in TCAB1 KO cells upon POT1dOB expression as data not shown, because it synergizes with telomere elongation in TCAB1 cells by HiT (Figure 6) to make the point that TCAB1 KO doesn’t preclude a cell from having long telomeres. We also liked the experiment because Coilin KO cells were not different from parental cells in telomere elongation by POT1 dOB, whereas they were different in telomere elongation by HiT. The problem comparing parental cells to TCAB1 KO cells with POT1 dOB expression is that the TCAB1 KO cells start out with shorter length. Net amount of telomere lengthening does appear less in TCAB1 KO than parental, but as a fold-increase from where they started, the two backgrounds are similar.